# The Executive Functioning Paradox in Substance Use Disorders

**DOI:** 10.3390/biomedicines10112728

**Published:** 2022-10-28

**Authors:** Louise Jakubiec, Valentine Chirokoff, Majd Abdallah, Ernesto Sanz-Arigita, Maud Dupuy, Joel Swendsen, Sylvie Berthoz, Fabien Gierski, Sarah Guionnet, David Misdrahi, Fuschia Serre, Marc Auriacombe, Melina Fatseas

**Affiliations:** 1University of Bordeaux, CNRS, INCIA, UMR 5287, 33000 Bordeaux, France; 2Department of Addictology, CHU Bordeaux, 33000 Bordeaux, France; 3Department of Addictology, CH Charles Perrens, 33000 Bordeaux, France; 4University of Bordeaux, CNRS, INCIA, UMR 5287, EPHE PSL Research University, 33000 Bordeaux, France; 5Parietal Team, INRIA Saclay Ile-de-France, 91120 Palaiseau, France; 6Institut Mutualiste Montsouris, Department of Psychiatry for Adolescents and Young Adults, 75014 Paris, France; 7Laboratoire Cognition, Santé et Société, University of Reims Champagne Ardenne, 51571 Reims, France; 8University of Bordeaux, CNRS, SANPSY, UMR 6033, 33000 Bordeaux, France

**Keywords:** ecological momentary assessment, substance use disorders, executive functions, craving, rsfMRI, resting state, relapse

## Abstract

Deficits in neurocognitive functioning are trait-like vulnerabilities that have been widely studied in persons with substance use disorders (SUD), but their role in the craving–use association and relapse vulnerability remains poorly understood. The main objectives of this study were to examine whether executive capacities moderate the magnitude of the craving–substance use relationship, and if this influence is correlated with the functional connectivity of cerebral networks, combining rsfMRI examinations and ecological momentary assessment (EMA). Eighty-six patients beginning outpatient treatment for alcohol, tobacco or cannabis addiction and 40 healthy controls completed neuropsychological tests followed by EMA to collect real-time data on craving. Fifty-four patients and 30 healthy controls also completed a resting-state fMRI before the EMA. Among the patients with SUD, better verbal fluency and resistance to interference capacities were associated with a greater propensity to use substances when the individual was experiencing craving. Preliminary rsfMRI results identified specific networks that interacted with executive performance capacities to influence the magnitude of the craving–use association. Individuals with better executive functioning may be more prone to relapse after craving episodes. Specifically, better resistance to interference and cognitive flexibility skills may reduce attention to distracting stimuli, leading to a greater awareness of craving and susceptibility to use substances.

## 1. Introduction

Substance use disorders (SUD) are defined by a loss of control over substance use, compulsive and continued use despite harmful consequences, and by craving that induces repetitive relapses [1,2]. The key role of craving in the relapse vulnerability and maintenance of SUD has been highlighted by numerous experimental, observational and daily life studies [3,4]. In particular, recent investigations using ecological momentary assessments (EMA) have documented the real-time, prospective link between craving episodes and subsequent substance use across various forms of SUD [2,5]. However, the extent to which individual traits or vulnerabilities may influence the relationship between craving and substance use remains poorly understood, and it constitutes an important barrier to the development of personalized treatment strategies.

Deficits in neurocognitive functioning are trait-like vulnerabilities that have been widely studied in persons with SUD [6]. Research in this area has focused particularly on the role of higher-order executive functions (e.g., attention, response inhibition, cognitive flexibility, working memory) that are necessary for planning, decision-making, goal-directed actions and self-regulation of impulsive behaviors [7]. Clinical and neurobiological studies have demonstrated major deficits in these executive functions (EFs) in individuals with SUD that interfere with these capacities across a range of substances [6,8,9,10]. EF deficits have also been associated with decreases in frontal cortex activity and other markers of brain function [11,12]. To date, however, the literature has largely focused on group differences in higher-order executive functions based on comparisons of persons with SUD relative to healthy controls, or on correlations of the executive performance scores of patients with global SUD outcomes. Experimental studies that have attempted to investigate the mechanisms implicated in relapse have failed to show consistent correlations across drug classes [13,14,15]. As a result, it is unclear if the observed correlations between EF deficits and clinical outcomes in SUD can be attributed to causal mechanisms or if such associations may be more complex than can be demonstrated by traditional research paradigms.

The application of EMA would allow for the craving–substance use relationship to be characterized in real time and as it naturally expresses itself in daily life, and the magnitude of such dynamic coefficients could be then examined relative to the individual’s specific neurocognitive profile. Moreover, recent research has highlighted the value of resting state functional connectivity (rsFC) as a potential predictive biomarker of clinical outcomes and relapse in SUD [16,17]. Both comprehensive reviews and meta-analyses have coincided in describing the changes in functional connectivity related to substance and behavioural addiction in the striatum (including caudate, putamen and globus pallidus), thalamus, insula and frontal cortex regions, and brainstem nuclei [18,19]. However, there is no consensus yet regarding the nature of these changes, with reports including hypo- and hyper-connectivity compared with control subjects. Beyond region-to-region connectivity analysis, networks involving these regions have shown changes in the resting state functional connectivity (rsFC) between SUD patients and healthy controls (executive control network, ECN; salience network, SN; default mode network, DMN; limbic and reward networks), which are associated with craving and subjective withdrawal (ECN, DMN, and reward and limbic networks) and treatment outcomes (ECN, SN). The combination of neuropsychological and EMA data with rsFC would therefore provide novel insights into the pathophysiology underlying the relationships among executive functioning, craving and relapse in SUD, eventually identifying the specific functional circuits associated with a higher relapse risk following craving episodes.

The present investigation examined this issue in outpatients with alcohol, tobacco, and cannabis use disorders. The main objectives were to (1) assess the independent associations of executive function with craving and substance use in daily life, (2) examine whether executive capacities moderate the magnitude of the craving–substance use relationship, and (3) explore whether this potential influence measured by EMA can be associated with changes in the brain’s functional connectivity.

## 2. Materials and Methods

### 2.1. Participants

In total, 126 individuals (86 patients with SUD and 40 healthy controls) participated in the study. Patients were recruited in the context of regular outpatient treatment for addiction and met the DSM-5 criteria for a current alcohol, tobacco, or cannabis use disorder. The patients received standard comprehensive care during the study, consisting of pharmacotherapy (when available) combined with individual behavioral treatment focused on relapse prevention and psychosocial support. Full abstinence was encouraged by physicians, but with no consequences for the patient if he or she failed to achieve this goal. Healthy control participants were identified through community postings and were recruited in the lifetime absence of psychotic disorder, bipolar disorder and substance use disorder, as well as no other current psychiatric diagnoses. All participants were also required to be free from conditions or disabilities incompatible with the use of a smartphone or any contraindication for an rsfMRI examination.

### 2.2. Procedure

After verification of the eligibility criteria, all participants provided written informed consent, completed a battery of clinical and neuropsychological assessments, and were trained to operate a study-dedicated smartphone (Samsung Galaxy S with a 10.6 cm screen, 12-point font size). Following successful completion of this training, they were then given a smartphone to carry with them for one week and were instructed to respond to five electronic surveys per day. The feasibility and validity of EMA has previously been demonstrated in substance use disorder patients [3,5]. The surveys occurred at random intervals within 5 equal time epochs from morning to evening (approximately every 3 h). The participants who also received an rsfMRI examination did so within 48 h before completing clinical testing and EMA (Figure 1). Financial compensation was provided, with a maximum of €100 in purchase vouchers for the completion of both the EMA and rsfMRI phases of the study. The study was approved by the institutional human research committee (No. 2014-A01668-39).

### 2.3. Clinical Measures

#### 2.3.1. Addiction and Psychiatric Data

DSM-5 diagnostic criteria for current substance use disorders were assessed using the Mini International Neuropsychiatric Interview, French Version 5.0.0 (MINI) [20]. Substance-related data were assessed using a validated French version of the Addiction Severity Index (ASI), modified to take tobacco addiction into account [21]. The Interviewer Severity Ratings (ISR) from the drug, alcohol and tobacco sections of the ASI were used to assess the severity of each participant’s addiction.

#### 2.3.2. Neuropsychological Assessments

The Stroop task [22] was administered to examine attention, interference and cognitive inhibition. This test is composed of three parts which correspond to three trial types: color trials (rectangles presented in different colors), word trials (names of colors printed in black ink) and incongruent trials (names of colors printed in incongruent color ink, e.g., the word “red” colored in blue). Each page contains 5 columns of 10 items. Participants were instructed to read the maximum number of words (word page) and name the maximum number of ink colors as quickly as possible, in 45 s. An interference score was calculated by subtracting a predicted color-word value from the obtained color-word score, with higher scores reflecting lower difficulties in inhibiting interference [23].

The Trail Making Test (TMT) measures cognitive and motor speed as well as mental flexibility and automated process inhibition [24]. Participants are asked to connect a series of circles with numbers in order in Part A and numbers alternating with letters in order in Part B, as quickly as possible. Part A of the TMT requires participants to sequentially connect numbered circles, while Part B requires participants to sequentially alternate between numbers and letters, and it is regarded to index executive functioning, specifically set-shifting flexibility, attention and inhibition. A lower difference score calculated by the difference in the completion time of the parts (B–A) reflects better executive functioning ability.

The Iowa Gambling Task (IGT) was administered to assess decision-making [25]. Participants were given an amount of money to start with and were told to maximize their profit over the course of 100 trials by selecting cards from one of four decks, among which two were advantageous and two disadvantageous. The net score was calculated by subtracting the total number of selections from disadvantageous decks from the total number of selections from advantageous decks.

The letter verbal fluency test was used to assess executive control ability, particularly inhibition, flexibility and updating ability. Participants were asked to produce as many words as possible beginning with a given letter (T and V) in one minute. The score was the sum of unique correct words (excluding repetitions, proper nouns and derived words) for each letter [26].

### 2.4. Ecological Momentary Assessment

At each electronic interview, the participants were prompted to rate the maximum level of craving to use substances that they had felt since the previous assessment on a seven-point scale from 1 (no desire to use) to 7 (extreme desire). They were also asked if they had used the substance that initiated their treatment since the last assessment, followed by questions concerning the use of any other substance during that time period (tobacco, alcohol, opiates, cocaine, amphetamine, cannabis or other substances).

### 2.5. Acquisition of Brain Imaging Data

Anatomical and functional brain imaging data were collected using a 3.0 Tesla GE MRI system with a 32-channel MRI head coil. Anatomical MRI volumes were acquired using a sagittal three-dimensional T1-weighted scan (repetition time, 8.5 ms; echo time, 3.2 ms; flip angle, 11°; FOV, 256 mm × 256 mm; voxel size, 1 mm × 1 mm × 1 mm; 176 slices). The resting-state functional MRI volumes were acquired using a single-shot echo-planar sequence (RT, 2200 ms; ET, 27 ms; flip angle, 80°; FOV, 192 mm × 192 mm; voxel size, 3 mm × 3 mm × 3.5 mm; 42 axial slices; number of volumes, 300). The total duration of the resting-state fMRI scan was 11 min, during which, the participants were instructed to keep their eyes closed, to not fall asleep and to not think about anything in particular.

#### 2.5.1. Data Preprocessing

Resting-state functional MRI and T1-weighted MRI images were preprocessed using fMRIPrep 20.2.1 [27,28] which is based on Nipype 1.5.1 [29,30]. Briefly, the T1-weighted structural scans underwent correction for intensity nonuniformity (INU), followed by skull stripping; brain tissue segmentation of the cerebrospinal fluid (CSF), white matter (WM) and gray matter (GM); and normalization to the Montreal Neurological Institute (MNI152NLin6Asym) space using nonlinear registration with ANTs. Preprocessing of the resting-state fMRI scans included slice-timing correction, susceptibility distortion correction, motion correction, co-registration to each subject’s preprocessed T1-weighted scan, normalization to the MNI152NLin6Asym standard space, resampling to a 2 × 2 × 2 mm grid and spatial smoothing (FWHM = 6 mm). Finally, we performed nuisance regression on the resting-state fMRI data by regressing out: (1) six aCompCor components from the WM and CSF separately; (2) 12 motion parameters representing three translation and three rotation time-courses and their temporal derivatives; (3) outlier volumes with a frame-to-frame displacement FD >0.5 mm, together with their temporal derivatives; and (4) linear and quadratic trends. While there were no significant between-group differences in terms of head motion, mean frame-to-frame displacement FD per subject was included as covariate in the group-level analysis to attenuate any remaining effects of head motion.

#### 2.5.2. Definition of Functional Connectivity at Two Levels of Topological Organization

Brain functional connectivity was analyzed at two topological levels, namely region-to-region connectivity and resting-state networks, to explore the feasibility of the combined analysis of the EMA and rsFC datasets. For the low-level connectivity analysis, we used the parcellation of the 512 finely resolved regions based on a recently introduced multi-scale functional parcellation (dictionaries of functional modes, DiFuMo). This scale of connectivity (the 512-dimensional DiFuMo) provides close results to analyses performed on signals at a higher resolution (the 1024-dimensional DiFuMo and voxel level) [31]. For each subject, we extracted blood-level dependent (BOLD) time-series from each of the 512 functional brain regions and estimated the functional connectivity between regions by computing the Pearson correlations between their respective BOLD time-series. This yielded a 512-by-512 matrix per subject, where the elements of the matrix represented whole-brain pairwise functional connectivity. In order to avoid artifactual anti-correlated connectivity resulting from the anatomical component correction method [32] we thresholded the functional connectivity matrices to retain only positive correlations and further removed those correlations including white matter regions to avoid misregistration of artifacts.

For higher-level topological connectivity analyses, we grouped the 512 regions’ parcellation into 17 distinct large-scale functional networks defined using the resting-state functional connectivity [33] (Appendix A DiFuMo512-2-Yeo17_dictionary). The parcellations of the 17 large-scale networks include executive control (ContA-C), default mode (DefaultA-C), dorsal attention (DorsAttn-B), salience/ventral attention (SalVentAttnA-B), limbic (LimbicA-B), somatomotor (SomMotA-B) and visual (VisCentral and VisPeripheral) networks. The functional connectivity value for each of the defined resting-state networks was defined as the arithmetic average of the retained positive functional connectivity values among the constituent regions of the salience/ventral attention network and between them and the regions of each of the other large-scale brain networks.

### 2.6. Statistical Analysis

ANOVA and chi-square tests were performed to compare the patients and controls on quantitative and qualitative variables, and to compare patients across substance groups. EMA and clinical data were analyzed using hierarchical linear and nonlinear modeling [34]. Data were time-lagged so that craving levels at any given assessment (T0) were used to predict substance use at the subsequent assessment on the same day (T1). All analyses were adjusted for the status of the T1 outcome variable as measured at the T0 assessment. The magnitude of these within-person coefficients were then examined as a function of the neuropsychological test scores, adjusting for age, sex, education, comorbidity and SUD subtype. For neuroimaging data, two permutation tests (50,000 permutations; network-based statistics NBS1.2) [35] were conducted between the functional connectivity matrices of the patient (all SUD subtypes combined) and control groups (contrasts: SUD > controls and SUD < controls) for both low-level (512 × 512 regions) and high-level (17 × 17 resting-state networks) topological connectivity.

## 3. Results

### 3.1. Sample Description

Table 1 presents the sociodemographic and clinical characteristics of the overall sample. The patients were older than the healthy controls and had fewer years of education, but the groups did not differ by sex. Concerning the neuropsychological tests, the SUD patients had poorer performance on the Stroop test and TMT than the healthy controls, but no differences were observed among the SUD subtypes. The mean number of years of substance use was 17.4 (16.26 years for patients with alcohol use disorder, 14.38 years for patients with cannabis use disorder and 20.53 years for patients with tobacco use disorder).

Adherence to the EMA methodology was high overall, with significantly greater compliance for healthy controls (95% of all administered assessments completed) as compared with patients with any form of addiction (85%), but with no differences among the SUD subtypes. Craving intensity was greater in the cannabis group relative to the alcohol group, and use of the substance at the origin of treatment was greater for the nicotine group compared with both other substance groups. The participants who underwent an rsfMRI examination (54 patients, 30 healthy controls) did not differ from those who participated only in the EMA phase of the study for any of the variables presented in Table 1, with the exception that the healthy controls who received an rsfMRI were more compliant with EMA than the controls who did not (96% versus 89%), and this group included proportionately more men than the EMA-only group.

### 3.2. Associations among Craving, Substance Use and Executive Functioning

The unadjusted average within-day association of craving with the use of any substance at the subsequent assessment (approximately three hours later) was significant among individuals with a SUD (γ = 0.158, SE = 0.031, *p* < 0.001) but not among healthy controls (γ = 1.448, SE = 0.722, *p* > 0.05). Craving was also strongly associated with the subsequent use of the substance that was the focus of treatment among SUD patients (γ = 0.181, SE = 0.036, *p* < 0.001) in the whole sample as well by substance group. This effect was greater in the alcohol group compared with the nicotine group for the use of any substance (γ = −0.167, SE = 0.070, *p* < 0.05) as well as the use of the treated substance (γ = −0.287, SE = 0.076, *p* < 0.001). The influence of each neuropsychological test score was then examined separately relative to craving and substance use, as well as concerning their association. After we adjusted for age, sex and education, no main effects were observed for the different tests relative to craving levels, the frequency of any substance use or the frequency of treated substance use. However, for patients with SUD, the within-person association between craving and later substance use was significantly modified by specific neuropsychological test scores. Specifically, better scores in the verbal fluency test were associated with an increased probability that craving would be followed by the use of any psychoactive substance (γ = 0.013, SE = 0.006, *p* < 0.05). As demonstrated in Table 2, this moderating effect was also observed for the Stroop test, whereby greater resistance to interference was associated with a stronger prospective association of craving with the use of any substance (γ = 0.003, SE = 0.001, *p* < 0.05), as well as with use of the substance necessitating treatment (γ = 0.004, SE = 0.002, *p* < 0.05). No other neuropsychological test score moderated the association of craving and substance use.

### 3.3. Altered Brain Connectivity Associated with Executive Performance, Craving and Substance Use

An analysis of differences in high-level functional connectivity between the patient and healthy control groups revealed no significant differences for any of the 17 resting-state networks. Group comparisons for lower-level functional connectivity (512 regions) showed that patients exhibited higher functional connectivity between the cuneus and globus pallidus, between the medial thalamus and posterior insula, and between the posterior thalamus and somato-motor region of the inferior central sulcus (Figure 2). In contrast, the control group demonstrated increased functional connectivity within the visual cortices (superior and inferior occipital gyrus), between the superior occipital gyrus and both the superior temporal gyrus and the posterior insula, and between the anterior putamen (anterior capsule limb) and the posterior corona radiata (superior longitudinal fasciculus).

The associations of these networks with EF scores were examined next by calculating the interactions of the Stroop and verbal fluency test scores with connectivity in each of the seven pairs of regions. The association of craving with the use of any substance varied as a function of the interaction of Stroop test performance and connectivity between the anterior putamen and the posterior corona radiata (γ = 0.009, SE = 0.003, *p* < 0.01), and connectivity between the superior occipital gyrus and the superior temporal gyrus (γ = −0.007, SE = 0.003, *p* < 0.05). The interaction of verbal fluency test performance with the connectivity between the superior occipital gyrus and the superior temporal gyrus significantly predicted the association between craving and any substance use (γ = −0.012, SE = 0.004, *p* < 0.01). For the association between craving and use of the substance necessitating treatment, significant interactions were observed for Stroop test performance with connectivity between the anterior putamen and the posterior corona radiata (γ = 0.009, SE = 0.004, *p* < 0.05). Finally, significant interactions were observed for verbal fluency test performance with connectivity between the anterior putamen and the posterior corona radiata (γ = −0.031, SE = 0.007, *p* < 0.001) in determining the magnitude of the association of craving with use of the treated substance.

## 4. Discussion

The present study examined whether executive functions moderate the dynamic, prospective association between craving and substance use among individuals with SUD, and whether such moderation may be linked to differences in functional brain connectivity. These questions were investigated through the joint analysis of data collected within the contexts of daily life using EMA along with data gathered through functional neuroimaging. Our results indicate an unexpected association between certain neuropsychological performance scores and the magnitude of the prospective link between craving and substance use in daily life. Specifically, better verbal fluency and resistance to interference capacities were both associated with a greater propensity to use substances when individuals were experiencing craving. Moreover, this finding was independent of the SUD type or additional comorbidity, suggesting that it may constitute a general process or phenomenon applicable to different forms of substance addiction. These EMA findings also underscore the seemingly paradoxical role that EF deficits may play in SUD. For example, while poor Stroop test performance has been associated with worse treatment retention [36] and tends to improve through treatment [37], many behavior studies have found mixed results in predicting clinical outcomes and relapse among treatment-seeking SUD samples [36,38,39]. EF in SUD may involve highly complex and, at times, opposite effects depending on the precise criterion examined.

The unexpected association we observed could be explained by several hypotheses. First, the findings suggest that patients with stronger abilities to resist interference might use those capacities to relieve craving more efficiently, including through focused substance-seeking behavior and substance use. By contrast, the elaborated intrusion theory [40] would indicate that this result represents a propensity for those patients to be less distractible, leading to enhanced salience or awareness of the craving experience, along with a lower capacity to distract themselves by stimuli from their environment (thereby leading to substance use). Should this hypothesis be valid, it is possible that the assessment of craving through EMA could have promoted thoughts and mental imagery related to substance use, without stimuli from the environment being able to prevent these thoughts. An argument in favor of this possibility is the observation of an interaction between executive performance and craving relative to connectivity between the occipital gyrus and the superior temporal gyrus, two areas involved in, respectively, mental/visual and verbal imagery. This assumption is also in line with the desire thinking model [41], which considers craving as a voluntary form of perseverative thinking underpinned by the implementation of imaginal (i.e., visual) prefiguration and verbal perseveration. In this way, patients with lower levels of EF impairment might have greater metacognition capacities that could lead to higher emotional distress and craving experience, and without sufficient daily life activities to reduce the desire to use substances. This explanation, however, is speculative and should be further investigated.

The results of neuroimaging demonstrated differences between the patients and controls that are in line with the existing literature describing the changes in functional connectivity related to substance and behavioural addiction [18]. While the preliminary nature and complexity of these findings prevent definitive conclusions, they appear to suggest a global pattern, whereby healthy controls have more direct connections between the somato-motor and visual regions, whereas connectivity in individuals with SUD often involves the striatal complex and the insula. Furthermore, in the patient group, several differences in connectivity were identified, which may help to explain how executive functions modify the craving–substance use association. Specifically, two networks (the anterior putamen and the posterior corona radiata, and the occipital gyrus and the superior temporal gyrus) interacted with executive performance capacities to influence the magnitude of the association between craving and substance use.

In contrast, our results did not find any association between impaired decision-making and the craving–use relationship, nor statistically significant differences between healthy controls and SUD patients regarding the IGT net score. This observation is inconsistent with common observations that decision-making capacities are impaired among patients with SUD, and with a review indicating a moderate to large association between decision-making capacities and relapse [6]. It is therefore notable that the studies in this review focused essentially on patients with a cocaine use disorder and relapse was defined as a return to use after abstinence, in contrast to our study, where the patients did not go through a mandatory abstinence period. In addition, this inconsistency could be explained by the possibility that cognitive deficits may mediate outcomes in a different manner than the craving–use associations, such as by interfering with the individual’s ability to effectively engage in treatment [42].

Several limitations of the present study should be considered in interpreting the findings. It should be noted that executive functions were assessed as stable individual traits, but we cannot exclude the fact that quick fluctuations in the capacity for inhibition in daily life may have an impact on the predictive value of craving episodes, contributing to further substance use. Furthermore, sex differences may modulate the interactions among neuropsychological characteristics, craving and substance use, and further studies should examine this issue. Concerning the neuroimaging data, the proof-of-concept approach combining EMA and rsFC datasets led us to adopt a conservative analytical strategy by removing all anti-correlations, thus focusing only on positively correlated networks, as these are the most consistently cited in the literature with regard to addiction [18]. This approach, in combination with the two permutation tests, one for each statistical contrast, provides a high level of confidence about the significant differences reported and a clear directionality for the resulting effects, but it does not deliver a complete account of more subtle differences and anti-correlated networks in particular.

Despite these limitations, our results have important clinical implications, pointing out the relevance of assessing individual differences in executive functions in order to identify individuals who may be more prone to relapse after craving episodes. The present work also argues for the need to develop additional targeted interventions aimed at helping individuals with a better capacity for resistance to interference to cope with craving. Moreover, knowledge of the neural circuits associated with executive functioning and the craving–substance use relationship also further our fundamental knowledge of the pathophysiology of SUD.

## Figures and Tables

**Figure 1 biomedicines-10-02728-f001:**
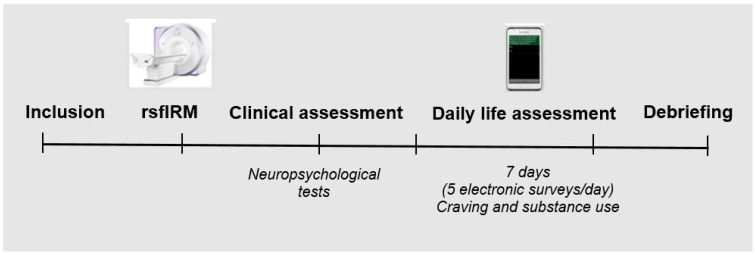
Overview of the method and procedures of the study.

**Figure 2 biomedicines-10-02728-f002:**
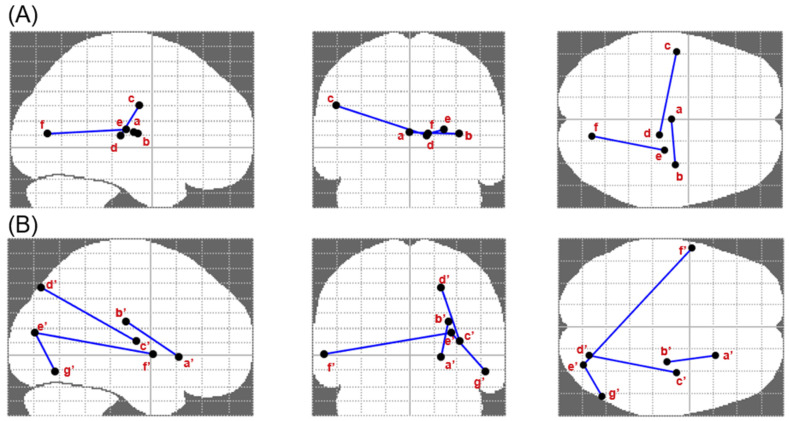
Comparisons of functional connectivity between the patient and healthy control groups. Significantly different functional connectivity between brain regions are displayed on the glass brain template (SPM–NBS) as links joining the corresponding functional nodes. (**A**) Patients with substance use disorders exhibited higher functional connectivity between the medial thalamus (a) and the posterior insula (b), higher functional connectivity between the somato-motor region of the inferior central sulcus (c) and the posterior thalamus (d), and higher functional connectivity between the globus pallidus (e) and the cuneus (f). (**B**) The control group demonstrated increased functional connectivity relative to the patients between the anterior putamen (anterior capsule limb, a′) and the posterior corona radiata (superior longitudinal fasciculus, b′), within the visual cortices (superior, e′, and inferior occipital gyrus, g′) and the contralateral superior temporal gyrus (f′), and between the superior occipital gyrus (superior part, d′) and the posterior insula (c′).

**Table 1 biomedicines-10-02728-t001:** Description of the sociodemographic, clinical and EMA variables of the sample.

	Healthy Controls (N = 40)		Any Addiction (N = 86)		Alcohol (N = 36)		Nicotine (N = 34)		Cannabis (N = 16)	
	M	SD %	M	SD %	M	SD %	M	SD %	M	SD %
Age	33.62	8.27	39.60	11.65 **	43.67	10.94 ^B^	38.82	11.84	32.13	8.99
Sex (% female)		50		43		36 ^A^		62		19
Education (years)	14.45	3.00	13.05	2.54 *	13.25	2.25	113.21	2.91	12.25	2.32
Addiction severity										
ISR			6.13	1.13	6.5	0.66 ^A^	5.65	1.37	6.31	1.08
Current comorbidity (%)										
Mood disorder		-		16		25 ^A^		6		19
Anxiety disorder		-		26		19 ^B^		18 ^C^		56
Psychotic disorder		-		20		8 ^A^		32		19
Any current	-			29		39		44		63
Neuropsychological tests										
Stroop interference	15.50	15.19	6.77	21.33 ***	7.78	30.29	6.70	12.57	4.68	9.02
TMT BA time	24.01	18.16	40.23	48.00 ***	38.23	34.97	46.70	66.91	31.00	12.63
IGT net score	12.35	28.72	10.22	25.08	11.91	27.30	5.24	21.82	17.20	25.87
Verbal/phonemic fluency	23.28	5.68	23.20	6.99	23.50	6.38	23.13	7.66	22.73	7.21
EMA										
Compliance	32.87	1.92	29.87	3.64 ***	30.61	2.96	29.88	3.22	28.19	5.26
Craving intensity	1.03	0.09	2.76	1.18	2.46	0.94 ^B^	2.73	1.22	3.49	1.34
Use of treated substance	-	-	15.83	10.24	10.36	8.34 ^A^	22.56	8.89 ^C^	13.81	8.89
Use of any substance	1.90	2.35	23.01	8.66	23.50	8.98 ^A^	23.35	8.67	21.19	8.16

* *p* < 0.05; ** *p* < 0.01; *** *p* < 0.001; ^A^ alcohol ≠ nicotine; ^B^ alcohol ≠ cannabis; ^C^ nicotine ≠ cannabis.

**Table 2 biomedicines-10-02728-t002:** Within-person associations of craving and use of any substance by resistance to interference (Stroop test).

Variable	Use of Any Substance					Use of Treated Substance				
	γ	SE	df	T Ratio	*p*	γ	SE	df	T Ratio	*p*
Unadjusted within-person association Craving/substance	0.158	0.031	78	5.069	<0.001	0.181	0.036	78	5.020	<0.001
Between-person moderators										
Age	−0.0003	0.003	78	−0.083	0.934	0.0002	0.003	78	0.061	0.951
Sex	−0.005	0.067	78	−0.073	0.942	−0.003	0.076	78	−0.040	0.968
Education	0.014	0.014	78	0.970	0.335	0.004	0.015	78	0.266	0.791
Nicotine (vs. alcohol)	−0.167 *	0.070	78	−2.422	0.018	−0.287 *	0.076	78	−3.790	<0.001
Cannabis (vs. alcohol)	−0.025	0.091	78	−0.271	0.787	−0.131	0.104	78	−1.259	0.212
Comorbidity	−0.056	0.077	78	−0.728	0.469	−0.041	0.080	78	0.535	0.594
Stroop test interference	0.003	0.001	78	2.225	0.029	0.004	0.002	78	2.462	0.016

* This negative value means that the alcohol SUD group was associated with a stronger prospective association between craving and the use of any substance or the treated substance than the nicotine SUD group.

## Data Availability

Not applicable.

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
