# Peer review of "The Executive Functioning Paradox in Substance Use Disorders"

_biomedicines, 2022, doi:10.3390/biomedicines10112728_

Round 1

Reviewer 1 Report

Authors  Jakubiec et al., have well written a manuscript titled "The Executive Functioning Paradox in Substance Use Disorders". The authors have presented the data analysis in a good manner. The authors have well-described study procedures in a detailed and easy way.

 Reviewer has minor concerns- 

1. Have authors found differences in executive performance based on the class of substance use such as alcohol, psychostimulants and opioids? 

2. It's better if authors would provide a graphical representation of the studies undertaken and method of analysis such as graphical abstract

3. Have the authors found any functional connectivity between substance abuse and frontal, temporal regions of the brain? 

Author Response

Point 1: Have authors found differences in executive performance based on the class of substance use such as alcohol, psychostimulants and opioids? 

Response 1: We thank the Reviewer for pointing this out. No significant differences in executive performances were found among the SUD subtypes. These results are presented in the Results Section and in Table 2.

Point 2: It's better if authors would provide a graphical representation of the studies undertaken and method of analysis such as graphical abstract

Response 2: We thank the Reviewer for the suggestion. For a better visualization of the methods, we added a graphical representation in the Methods Section (see Figure 1).

Point 3: Have the authors found any functional connectivity between substance abuse and frontal, temporal regions of the brain? 

Response 3: As indicated in our results (p. 7), patients exhibited higher functional connectivity between the cuneus and globus pallidus, the medial thalamus and posterior insula, and the posterior thalamus and somato-motor region of the inferior central sulcus. In contrast healthy controls demonstrated increased functional connectivity within the visual cortices (superior and inferior occipital gyrus), between the superior occipital gyrus and both the superior temporal gyrus and the posterior insula, and between the anterior putamen (anterior capsule limb) and the posterior corona radiata (superior longitudinal fasciculus). Even if the strength of connectivity within these connections, including frontal and temporal areas, were not regressed with substance use directly, they allowed to segregate patients from controls and thus may be involved in substance abuse. Furthermore few of these connections did impact the links between craving and substance use, as indicated p.7.

Author Response

-Major points:

Point 1: Figure 1 reports the use of the SPM glass brain template. Hence, since the authors used SPM: is it possible to assess the relation between rsfMRI patterns and executive functioning through a whole-brain voxel-wise analysis?

Response 1: We thank the reviewer for this interesting suggestion. Indeed, the imaging analyses described in our paper include the study of the brain’s functional connectivity characteristics at 2, well-differentiated levels of complexity. The higher-level analysis focuses on 17 resting-state networks as defined by Yeo et. al. (Yeo et al.,  2011). The lower-level analysis is based on the DiFuMo (Dictionaries of Functional Modes) (Dadi et al., 2020) multi-scale template, of which we chose to use the template defined by 512 regions (https://parietal-inria.github.io/DiFuMo/512 ). This template’s regions are defined by a functional decomposition model (dictionary learning; Olshausen and Field, 1997) which should better account for behavioural outcomes than purely anatomically-based parcellations since it is based on the funtional structure of the brain. 

In contrast, voxel-level functional connectomes have a number of disadvantages: most importantly they are computationally and statistically intractable as working at this level of detail entails modeling hundreds of millions of connections. Furthermore, in order to account for the multiple comparison problem generated, stringent corrections to the statistical analysis must be applied. Thus, the average of signals on regions or networks illustrated by our approach constitutes the standard analytical strategy. To our knowledge, the best example of a power comparison between analyses at voxel-level versus higher-scale anatomical or functional parcellations is provided by Dadi et al. (2020). In their paper, the authors convincingly conclude that “standard task-fMRI analysis on signals derived from 512 or 1024-dimensional DiFuMo gives results close to the voxel-level gold standard” while the dimensionality reduction involved has the “additional benefit of alleviating the burden of correcting for multiple comparisons” and further easying computational demands.

We have now clarified this in the manuscript in the Methods section by adding the following sentence (Section 2.5.2):

“For the low-level connectivity analysis, we used the finely-resolved 512 regions parcellation based on a recently introduced multi-scale functional parcellation (Dictionaries of Functional Modes, DiFuMo). This scale of connectivity (512-dimensional DiFuMo) provides close results to analysis performed on signals at higher resolution (1024-dimensional DiFuMo and voxel-level) (Dadi et al. 2020).”

Kamalaker Dadi, K., Gaël Varoquaux,G., Antonia Machlouzarides-Shalit,A., Krzysztof J. Gorgolewski, KJ.,Demian Wassermann, D.,Bertrand Thirion,B., Arthur Mensch,A. Fine-grain atlases of functional modes for fMRI analysis, NeuroImage, Volume 221, 2020,117126,ISSN10538119, https://doi.org/10.1016/j.neuroimage.2020.117126.

Point 2: The current contrast is based on the pooled SUD patient group vs controls. Since the data show some variance in executive functioning (EF) within SUD patients, can patients be split in: low EF, moderate EF and high EF groups, so that the alterations in rsfMRI networks could be visualized within these three different groups, contrasted to controls or even to each other? Splitting scores should be done in a data-driven way, e.g. by R functions.

Response 2: We thank the reviewer for this comment. Although it would be indeed interesting to analyze the patients subgroup differences in functional connectivity, the low number of subjects per group of each subdivision of the patient group (54 subjects) would render the statistical analysis impossible due to lack of statistical power.

Point 3: There are better ways to report network connectivity: e.g. in which the line thickness and node shape corresponds to the strength of connectivity (Brain Connectivity Toolbox running on Matlab)

Response 3: We thank the reviewer for this suggestion. Indeed, when the network under analysis is extensive, comprising dozens if not hundreds of connected nodes, the visualization rapidly becomes complex and benefits from additional information to enhance the salience of particular sets of connections that should be outstanding. However, the low number of connections of the resulting network in this study is not high enough to justify the addition of connectivity strength to anatomical information that the graph provide.

- Minor points

Point 1: Methods: p5: 17 large-scale networks were used: can the individual ROIs, belonging to each of these networks be added to the manuscript (in (Suppl) Table)?

Response 1: We thank the reviewer for this suggestion. We agree that providing the correspondence between both segmentations can increase the readability of the study. Thus, we have incorporated the table to the supplementary material (DiFuMo512-2-Yeo17_dictionary). We have also edited the text accordingly (Section 2.5.2.):

“For higher-level topological connectivity analyses, we grouped the 512 regions parcellation into 17 distinct large-scale functional networks defined using resting-state functional connectivity [33] (supplementary table DiFuMo512-2-Yeo17_dictionary).”

The original correspondance between DiFuMo-512 and the Yeo-7 and -17 segmentations can be found online at: https://github.com/Parietal-INRIA/DiFuMo/blob/master/difumo_labels/labels_512_dictionary.csv

Point 2The methods mention that the severity of addiction was assessed using ISR. However, table 1 does not report these metrics. I wonder if there would be an effect on the EF outcome moderation effect when ISR was considered?

Response 2: We apologize for this missing result. The ISR values have been now presented in the Table 1. No moderating effect was observed for the ISR concerning the within-person association between craving and later substance use (use of the treated substance (p=0.874) or any substance (p=0.058)) when considering the EF outcome.

Point 3: Table 2 reports gamma values: can the authors explain what a negative value means: -0.155: does this mean alcohol SUD patients > nicotine SUD patients?

Response 3: This negative value means that alcohol SUD group was associated with a stronger prospective association of craving with the use of any substance compared to the nicotine SUD group.

Point 4: Table 2 should report the exact p-values in a separate column instead of *.

Response 4: This has been done.

Point 5: Change “MRI” throughout the text a more detailed reporting of the actualsequence/modality of MRI which is “rsfMRI

Response 5: This has been done throughout the text.

Round 2

Reviewer 2 Report

The authors have not answered point 2 and 3.

I understand there might be a power issue with n=54, but then at least execute the proposed analysis and mention the obtained group sizes after data-driven splitting based on EF score. It is not unusual for imaging studies to have groups of ~ 20 cases.

Moreover, what happened with the values in Table 2? The gamma values have changed compared to the previous version. The meaning of negative gamma values should be explained in the legend of the Table to facilitate interpretation for the reader (not only explained to the reviewer).

Implementing my suggestion for point 3 could improve the data visualization and interpretation as well.
